# Experimental evolution of a pathogen confronted with innate immune memory increases variation in virulence

Ana Korša[1]*, Moritz Baur[1], Nora K. E. Schulz[1], Jaime M. Anaya-Rojas[1], Alexander Mellmann[2], Joachim Kurtz[1]*

**1** Institute for Evolution and Biodiversity, University of Münster, Münster, Germany, **2** Institute for Hygiene, University of Münster, Münster, Germany

☉ These authors contributed equally to this work.
* joachim.kurtz@uni-muenster.de (JK); korsa@uni-muenster.de (AK)

## Abstract

Understanding the drivers and mechanisms of virulence evolution is still a major goal of evolutionary biologists and epidemiologists. Theory predicts that the way virulence evolves depends on the balance between the benefits and costs it provides to pathogen fitness. Additionally, host responses to infections, such as resistance or tolerance, play a critical role in shaping virulence evolution. But, while the evolution of pathogens has been traditionally studied under the selection pressure of host adaptive immunity, less is known about their evolution when confronted to simpler and less effective forms of immunity such as immune priming. In this study, we used a well-established insect model for immune priming – red flour beetles and their bacterial pathogen *Bacillus thuringiensis tenebrionis* – to test how this form of innate immune memory drives the pathogen evolution. Through controlled experimental evolution of the pathogen in primed versus non-primed hosts, we found no change in average virulence after eight selection cycles in primed host. Nonetheless, we observed a notable rise in the variability of virulence, defined as the ability to kill hosts, among independent pathogen lines that evolved in primed hosts, and the bacteria were unable to develop resistance to host priming. Whole genome sequencing revealed increased activity in the bacterial mobilome (prophages and plasmids). Expression of the Cry toxin – a well-known virulence factor – was linked to evolved differences in copy number variation of the *cry*-carrying plasmid, though this did not correlate directly with virulence. These findings highlight that innate immune memory can drive variability in pathogen traits, which may favor adaptation to variable environments. This underscores the need to consider pathogen evolution in response to innate immune memory when applying these mechanisms in medicine, aquaculture, pest control, and insect mass production.

**Data availability statement:** The genomic sequences can be found on NCBI under Bioproject number: PRJNA1174726. All the data sets, code, and RMarkdown file with figures and analysis results can be found in Zenodo depository: https://doi.org/10.5281/zenodo.15050213

**Funding:** This work was supported by the Deutsche Forschungsgemeinschaft (DFG, German Research Foundation) within the Research Training Group GRK 2220 "Evolutionary Processes in Adaptation and Disease," project number 281125614 to JK and partial funding within SFB TRR 212 (NC³) – Project numbers 316099922 and 396780003 (to JK). The funders had no role in study design, data collection and analysis, decision to publish, or preparation of the manuscript.

**Competing interests:** The authors have declared that no competing interests exist.

## Author summary

Understanding pathogen evolution is crucial for science and public health. A key question is whether pathogens evolve variations in virulence, which quantifies the harm inflicted on the host. While research often focuses on pathogens confronted with their hosts' adaptive immune system, less is known about pathogen evolution in response to simpler kinds of immunity. Immune priming is a form of memory in the innate immune system of invertebrates. In this study we used red flour beetles and their bacterial pathogen *Bacillus thuringiensis tenebrionis* as a model to explore immune priming's influence on pathogen evolution. After eight infection cycles in primed or control hosts, pathogens in primed hosts showed higher virulence variability compared to controls. Bacteria did not develop resistance to immune priming, indicating response robustness. Genetic analysis revealed increased activity in mobile genetic elements, including prophages and plasmids, with variations in a virulence-related plasmid encoding the Cry toxin. These findings suggest innate immune memory can promote diversity in pathogen traits, influencing adaptation in unpredictable environments. Our results highlight the importance of considering pathogen evolution in response to innate immune memory, especially when applying immune priming strategies in medicine, agriculture, and pest control.

## Introduction

Virulence is usually defined as the ability of a pathogen to reduce host fitness (or its components, e.g., survival during the infection process [1,2]. The direction in which virulence evolves in an evolutionary arms race often depends on the balance between the benefits and costs it provides to the pathogen. For instance, when the highly virulent *Myxoma* virus was introduced into Australian rabbit populations in the 1950s, it evolved to intermediate levels of virulence while maximizing its fitness [3–5]. In contrast to highly virulent strains that rapidly kill hosts—potentially reducing transmission opportunities—intermediate virulence allows for sustained infection and transmission while still overcoming host defences. In this example, the cost associated with high virulence was host death, which limited pathogen transmission. Although transmission of pathogens varies and some pathogens can only be transmitted after host death, several studies have shown that optimal transmission rates happen at intermediate levels of virulence [6–8]. Additionally, several experimental evolution studies have revealed the complexity of selection processes acting on both hosts and pathogens during their rapid coadaptation [9,10] in which the coevolution between the host and pathogen favours and maintains intermediate levels of pathogen virulence [10].

Host responses and defence mechanisms against pathogens also play a crucial role in the evolution of virulence. Depending on the ability of organisms to resist or tolerate infections, pathogen virulence evolution can take different trajectories [11]. The evolution of tolerance, for instance, can strongly influence the evolution

of virulence by allowing pathogens to evolve higher transmission rates without increasing the actual level of virulence in tolerant hosts. Consequently, pathogens can develop extreme virulence in hosts lacking tolerance, potentially leading to devastating epidemics. This occurs because non-tolerant hosts are incapable of reducing the damage caused by high pathogen burdens, which results in elevated mortality rates [11]. Likewise, epidemiological and mathematical studies propose that "leaky" vaccines that do not offer complete immunity might allow pathogens to continue spreading and developing greater virulence, thus making them more dangerous to unvaccinated individuals [3,12–14]. On the other hand, leaky vaccines can also lead to reduced pathogen virulence through the reduction of pathogen load [15].

Virulence evolution is a complex trait that may also be influenced by interactions with the hosts' microbiome [16]. A recent experimental evolution study in nematodes and the pathogen *Pseudomonas aeruginosa* revealed that incomplete host immunity that allows infection with reduced disease severity, caused by host microbiota, favours the evolution of higher pathogen virulence [17]. Moreover, it has been demonstrated that the external immune defences of red flour beetles, specifically their antimicrobial quinone-rich secretions, can influence the evolution of pathogens, leading to increased virulence in pathogens like *Metarhizium brunneum* [18]. On the other hand, the presence of defensive microbes within the host, as demonstrated in *Caenorhabditis elegans* infected with *Serratia marcescens*, can lead to the evolution of reduced pathogen virulence by limiting pathogen success and transmission [19]. The above examples illustrate how host immunity can influence the evolutionary trajectories of pathogen virulence in different ways. More empirical data from experimental evolution studies across diverse host-pathogen systems are necessary to gain a broader understanding of the conditions under which host immunity promotes higher or lower pathogen virulence.

Invertebrate innate immune priming has been suggested to be analogous to leaky vaccination, as primed individuals can still become infected but also can tolerate elevated pathogen loads, providing an experimental system to assess the effect of innate immune memory on pathogen evolution [20]. Immune priming is a form of immune memory provided by the innate immune system, where initial non–lethal pathogen exposure can lead to enhanced protection upon secondary exposure [21–23]. Over the last couple of decades, the body of evidence for immune priming in invertebrates has been constantly growing [24–26] and experimental evolution studies revealed the rapid evolution of immune priming [27,28]. Innate immune systems of vertebrates show similar phenomena, in the form of trained immunity [29,30]. The relevance and potential applications of these forms of innate immune memory are currently intensively discussed [31–33]. While it has been observed that priming reduces pathogen transmission [34], we still, however, lack empirical data regarding the effects of these innate forms of immune memory on pathogen evolution and virulence.

In this study, we used the red flour beetle *Tribolium castaneum* and its natural pathogen *Bacillus thuringiensis tenebrionis* (*Btt*), to investigate how host priming influences the evolutionary trajectories of pathogen virulence. *T. castaneum* is a well-established model for host-pathogen coevolution and immune priming [35–37] and shows a specific priming response towards the spore-forming pathogen *Btt* after both septic and oral exposure [38–40]. *Bacillus thuringiensis (Bt)*, widespread and commonly present in diverse environments, produces insecticidal crystal proteins known as Cry toxins, specifically toxic to different insect orders [41]. Insects usually ingest *Bt* spores while feeding on contaminated food sources. After the spores are ingested orally, *Bt* can infect and reproduce within beetle larvae, ultimately leading to host death [42,43]. Upon host death, *Bt* undergoes sporulation inside the cadaver, producing new spores that are subsequently released into the environment. In nature, other hosts can encounter these spores either by scavenging infected cadavers or through environmental contamination. The repeated exposure of *T. castaneum* larvae to *Btt* spores in nature suggests that immune priming may be ecologically relevant, potentially influencing both host resistance and pathogen evolution over time. The oral priming response in our lab is triggered by the ingestion of the sterile filtered supernatant of the spore culture [39], which induces the upregulation of immune genes [44]. For example, previous research has shown that recognition genes and reactive oxygen species (ROS)-related genes are upregulated following oral priming with *Btt* suggesting heightened state of immune "alertness" characterized by increased expression of recognition genes and elevated ROS-based defences. In contrast, antimicrobial peptide genes (AMP) genes and genes involved in cellular responses were shown to be only up-regulated upon *Btt* infection and not priming [44].

To investigate how host immune priming influences the evolution of pathogen virulence, we experimentally evolved *Btt* in primed and non-primed beetle larvae for eight evolution cycles (Fig 1). During the experiment, beetles were not allowed to evolve by taking them from a stock population (i.e., one-sided evolution of the pathogen). After the eight cycles of experimental evolution, we compared the levels of virulence (proportion of mortality) and transmission (number of spores produced in cadavers of evolved *Btt* lines in primed and control host environment). We also sequenced the genomes of evolved *Btt* to identify whether genetic changes contributed to evolved differences in virulence. To our knowledge, our study is the first to show the influence of insect specific immune priming on pathogen evolution, thus broadening our understanding of how host innate immune memory might influence the trajectories of virulence evolution.

## Results

### Does host immune priming affect pathogen virulence evolution?

*B. thuringiensis tenebrionis (Btt)* bacteria evolved experimentally in either immune primed or control (i.e., non-primed) *T. castaneum* hosts (Fig 1, left side). After this selection treatment for eight cycles, we analysed the response of *Btt* to the

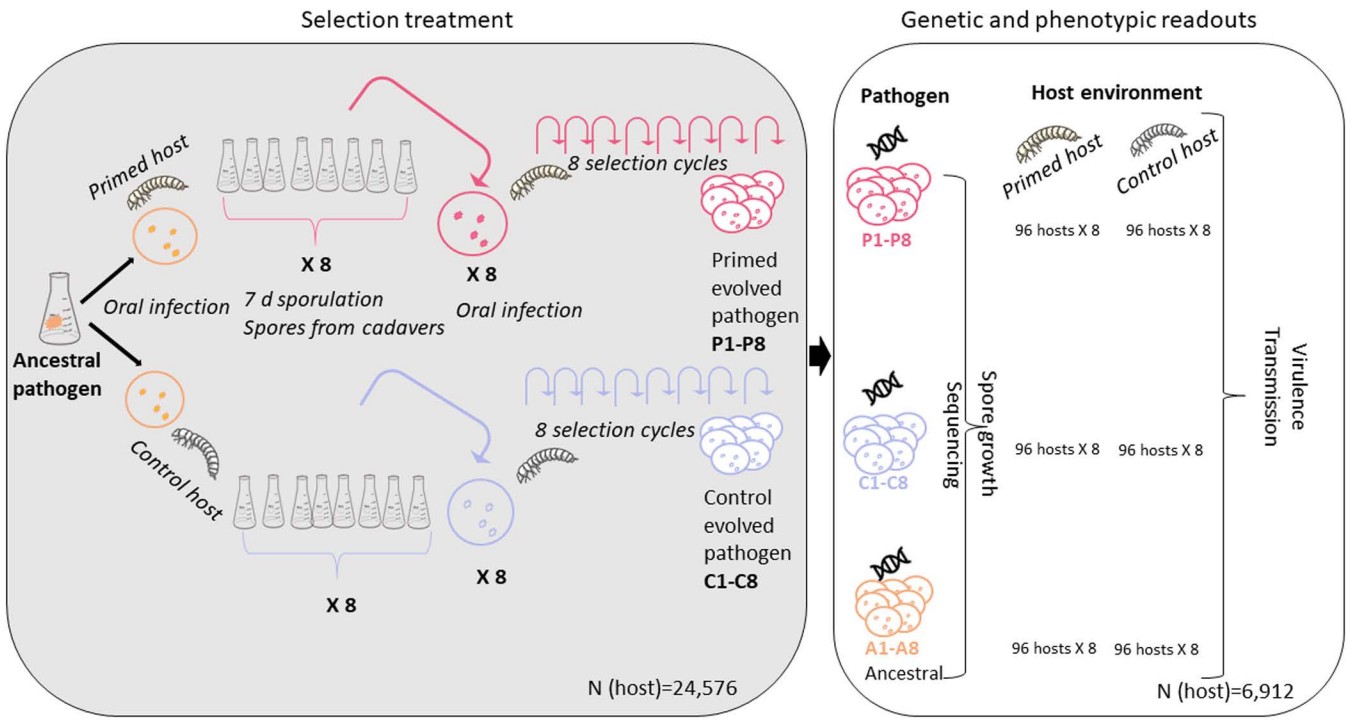

**Fig 1. Experimental design.** We inoculated one colony of the pathogen *Bacillus thuringiensis tenebrionis* (*Btt*) into 100 mL of *Btt* medium. After 7 days of sporulation, we harvested the spores, adjusted the concentration to 5x10^9, and produced an infection diet that was offered to individual primed and control 15 days old beetle larvae (n = 192 per replicate line and cycle). For each selection cycle, we used *Btt* spores from cadavers of *T. castaneum* larvae killed by the infection in the previous cycle and produced eight bacterial lines per selection treatment. For each bacterial line, we isolated spores from five cadavers, which were subsequently grown independently in a sporulation medium to achieve high spore numbers for the infection of a sufficient number of beetle larvae for the next infection cycle. In each selection cycle, bacteria reproduced and sporulated within their host followed by amplification in culture medium. We had eight independently evolving pathogen replicate lines each for evolution in control and primed hosts. After eight selection cycles, we phenotyped and genotyped all evolved pathogen lines (P1-8, C1-8) and the ancestral pathogen. For the ancestral pathogen, we produced pseudo-replicates (A1-8), i.e., lines that had not undergone any selection process, but had grown in separate flasks to control for possible variations from growth in the medium. For phenotyping, we quantified virulence (proportion of host mortality) and transmission (spore growth in cadavers) in two different host environments (primed and control). We also measured spore growth in a liquid medium. For genotyping, we sequenced the whole genomes of the 24 lines and quantified phage and plasmid coverage. We also measured the expression of a key virulence factor, the Cry-encoding gene, with RT-qPCR. Insect drawings by Helle Jensen.

selection pressure encountered in primed *vs.* control hosts. Previous research has indicated that *Btt* completes roughly 9.5 generations within the host [43], leading us to estimate that our experiment involved approximately 76 generations in the host. We then compared the virulence (measured as host mortality) of eight primed-evolved *Btt* lines (P1-8) and eight control-evolved *Btt* lines (C1-8) as well as ancestral *Btt* (eight replicates directly taken from stock *Btt*, A1-8). These phenotypic readouts (Fig 1, right side) were taken from two environments, primed and control host, thereby establishing a 'common-garden' experiment.

We found that both, the selection treatment (Fig 1, left side) and the host environment (Fig 1, right side) significantly affected pathogen virulence (Fig 2A; glmer (mortality~ selection_treatment*host environment) +(1|Replicate_line)+ (1|Block)+(1|Plate): $X^2 = 74.25$, Df = 5, p < 0.001). Selection in primed and control hosts significantly reduced the virulence, compared to the ancestral pathogen (control: p = 0.001; primed: p = 0.006). Furthermore, virulence of evolved lines was lower in the primed compared to control host environment (p < 0.001). This shows that priming was still efficient, also against the lines that had evolved in the primed hosts (Fig 2A).

We then performed separate analyses for primed and control host environments to see how individual bacterial lines vary in virulence in different host environments. To do that we modelled the effects of the selection treatments using a Beta distribution in a Bayesian framework and allowed the selection treatments to influence the precision parameter (ϕ). The mortality of control hosts infected with primed-evolved bacteria had an 85.15% probability of being higher than when infected with bacteria from the control pathogen lines, though the estimated difference remained small (6.6%, 95% CI: [-5.5, 18.7], see Fig 2B). In the primed host environment (Fig 2C), we found a similar pattern, with a 59.87% probability of higher mortality when primed hosts were infected with bacteria from the primed lines compared to when they were infected with bacteria from the control lines. However, the estimated difference was minimal (1.4%, 95% HDI: [-8.1, 12.3%]).

In terms of relative variability (i.e., coefficient of variation, k-CV), primed-evolved bacteria had a 96.6% probability of higher variability in virulence than control-evolved bacteria when infecting the control host (Fig 2D) and there was a 12.2% average increase in variability in virulence in primed-evolved bacteria (95% HDI: [-1.5, 27.6]). The relative variability in virulence in primed-evolved pathogens was not different from the variability of control-evolved bacteria when in primed host environment (1.8%, 95% HDI: [-15.3, 10.4], Fig 2E). However, the magnitude of variation in virulence was much higher in primed hosts than in control hosts.

### Do the evolved pathogens differ in spore production, sensitivity to priming and fitness?

To evaluate the spore production of the evolved bacteria, we measured the pathogen spore load in cadavers and the spore production in liquid medium. As *Btt* likely transmits through spores produced in cadavers [45], we considered the spore load a good proxy of the transmission rate. There was no difference in spore production between selection treatments in the control host (Fig 3A; lmer: $X^2 = 0.859$, Df = 2, p = 0.650), but in primed hosts, pathogens that had evolved in primed and control hosts produced fewer spores than the ancestral *Btt* (Fig 3B; lmer: $X^2 = 8.807$, Df = 2, p = 0.0122; primed-evolved: p = 0.014, control-evolved: p = 0.020). We also examined the correlation between spore load and virulence for all bacteria lines and selection treatments in the primed and control host environments but found no correlation (Figs 3C, S1 and S2). Finally, selection treatment had a significant effect on spore production in a liquid medium (F = 40.087, Df = 2, p < 0.001), where both primed-evolved and control-evolved pathogens exhibited lower spore production than the ancestral *Btt* (Fig 3D).

Additionally, we calculated the priming sensitivity of bacterial replicates by dividing virulence in control hosts by virulence in primed hosts. This gave us a value indicating how sensitive the pathogen is to the priming response of the host (Fig 4A). The more sensitive to priming the pathogen is, the lower the virulence in the primed hosts would be. Selection treatment showed a trend in affecting the priming sensitivity ($X^2 = 5.985$, Df = 2, p = 0.050), with the primed-evolved pathogen exhibiting a trend (p = 0.058) toward higher priming sensitivity compared to the ancestral pathogen (Fig 4A).

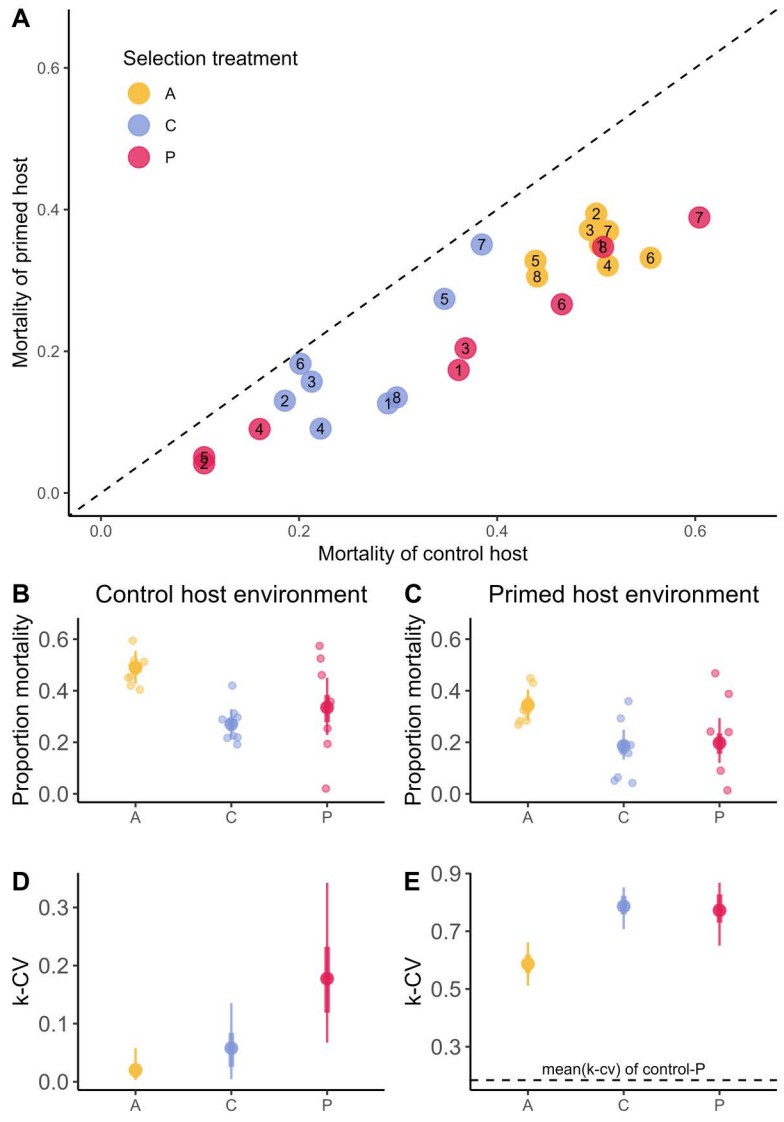

**Fig 2. Virulence (proportion of host mortality) of pathogen lines** **A**. Correlation between the virulence in primed and control hosts, showing raw data of each independent replicate in primed and control host environment larvae. The dotted line shows where the mortality rates of control and primed hosts would be equal. All pathogen lines are below this line, indicating lower pathogen virulence in primed compared to control hosts (i.e., a 'priming effect'). In panels **B to E**, large solid circles represent the posterior mean of the Bayesian model, and the smaller circles the raw data corresponding to the independent pathogen replicate line. Additionally, the thick and thin bars represent the 68% and 95% HDI, equivalent to a normal distribution's 1 and 2 standard deviations. **Panels B and C** show the virulence in control and primed host environments. **Panels D and E** show posterior estimates of k-CV, which is a measure of dispersion, calculated from the coefficient of variation. Points show the median estimates, and error bars represent the credible intervals (HDCI), indicating uncertainty.

We also assessed the fitness of each bacterial line within the host by combining mortality rates and spore load data (Fig 4B). Our analysis revealed that both selection treatments and host environment significantly influenced pathogen fitness ($X^2 = 40.385$, df = 5, p < 0.001). The primed host environment showed a significant negative effect on pathogen fitness (Estimate = -1.03, p = 0.0012), indicating that the within-host fitness in the priming environment is lower than in the control environment. We finally tested whether priming sensitivity was correlated with pathogen fitness, and found a

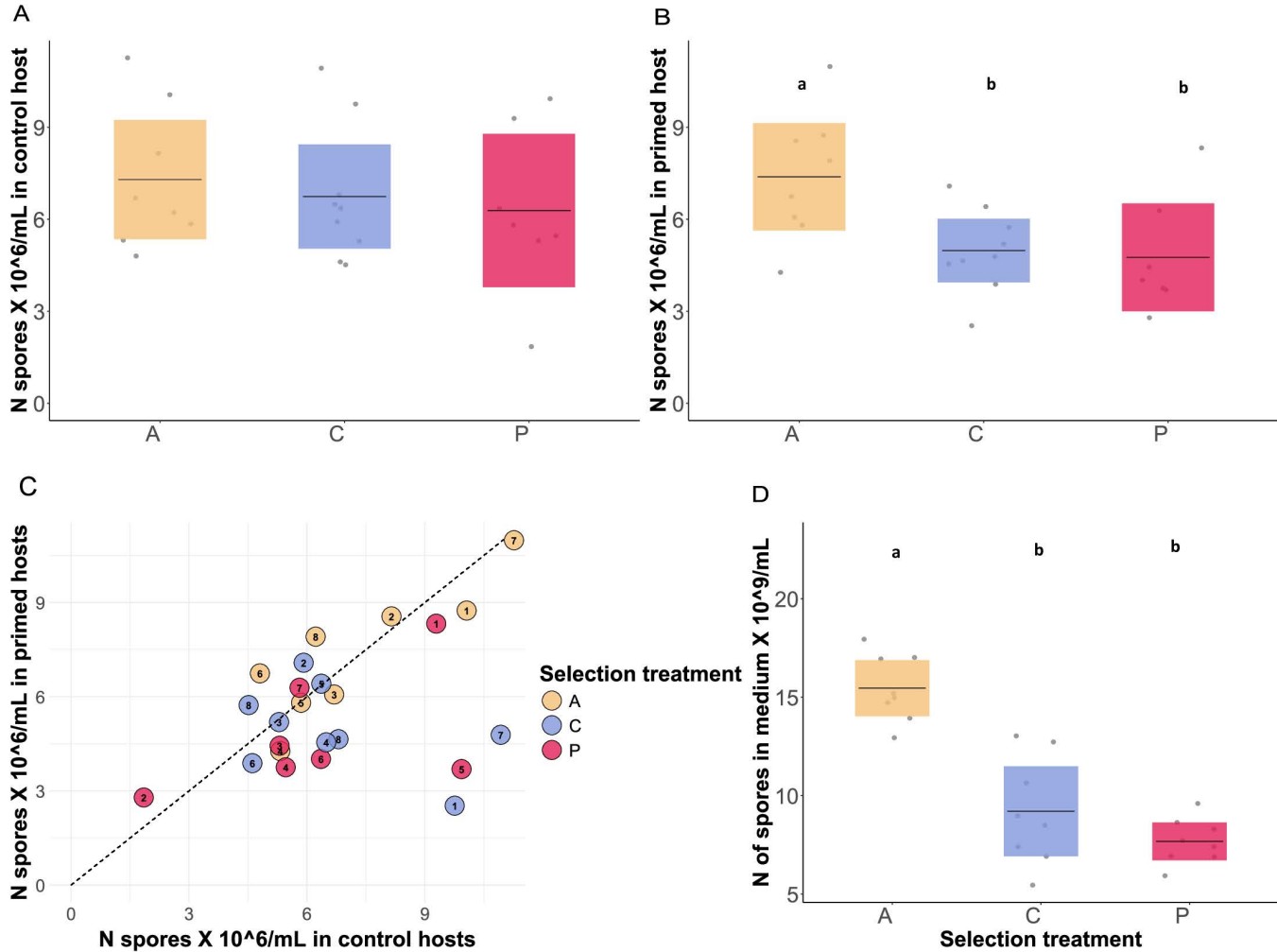

**Fig 3. Spore growth in A) primed and B) control hosts. The dots correspond to the mean of ten measurements for each independent pathogen line.** The plot shows the mean and the confidence intervals. Ancestral *Btt* produced more spores than both evolved lines, but only in the primed host environment (lmer: $X^2 = 8.808$, Df = 2, p = 0.012, and there seemed to be a weak difference in pathogen load between the evolved lines (p = 0.973). Letters denote significant differences between the selection treatments. **C.** Correlation between the spore load in primed and control hosts, showing the number of spores each independent replicate produces in primed and control beetle cadaver. The dotted line shows where the spore load in control hosts and primed hosts would be equal. **D** Spore growth in the medium. Each dot represents the mean of three independent measures of one pathogen line. The graph shows the mean and confidence intervals. Primed- and control-evolved lines produced fewer spores than the ancestral strain (p < 0.001).

negative trend for the primed-evolved pathogens (Fig 4C). Pathogens with higher fitness seem to be less impacted by the host's primed immune response. This suggests that these fitter strains may have evolved strategies to evade or resist the immunological effects associated with priming.

### Do the evolved pathogens differ in *Cry* gene expression?

Cry is an important virulence factor of *Bt*, but it can be costly, leading some bacteria to potentially lose it [10,41,46,47]. We tested for potential differences in *cry3a* gene expression, as evolved differences in virulence might be related to this important virulence factor. For this, we performed qPCR on RNA extracted from overnight cultures. In contrast to the negative control (*Btt* without *Cry3a*), all tested lines expressed *Cry3a* (S2 Table). To comprehensively assess these

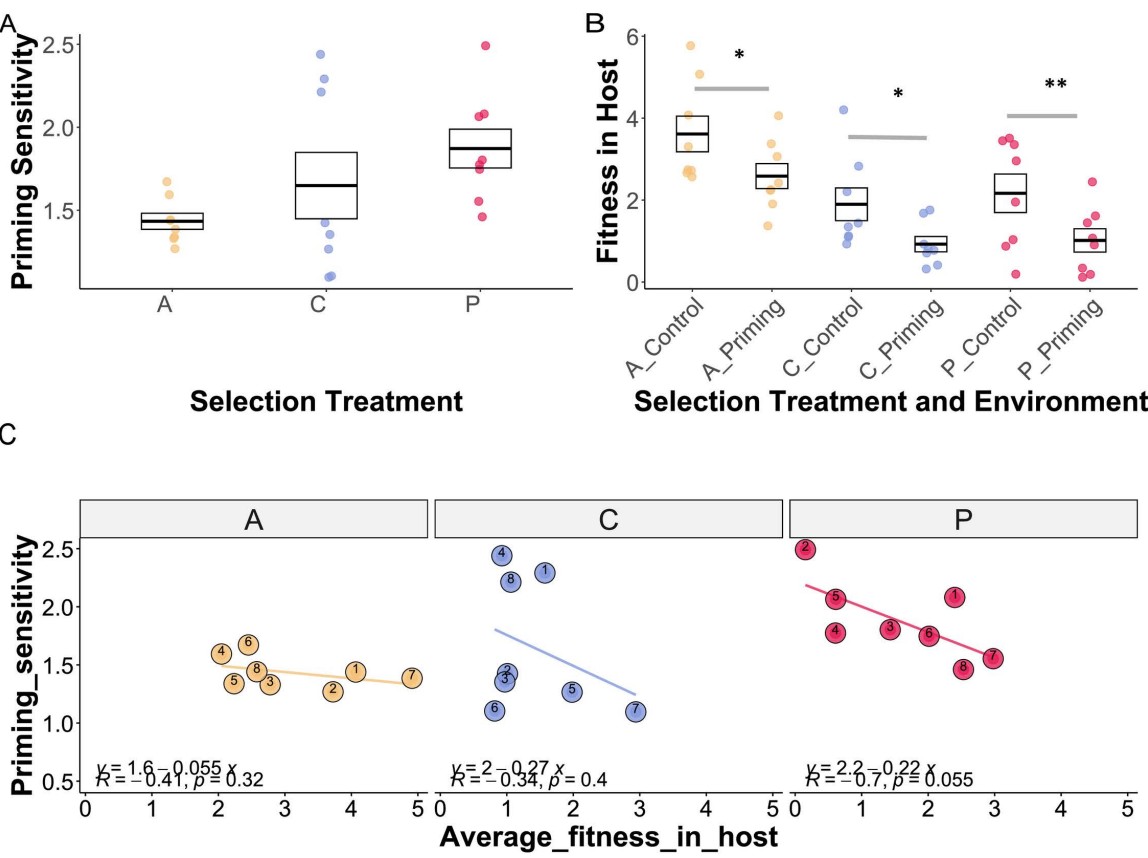

**Fig 4. Priming sensitivity and fitness in the host.** Each dot corresponds to bacterial replicate line and the box shows mean and confidence intervals. **A** Priming sensitivity of bacterial replicates was calculated by dividing the *proportion of mortality in control/the proportion of mortality in primed hosts*. The higher the value, the more sensitive the pathogen is towards host priming. Selection treatment affected the priming sensitivity ($X^2 = 5.985$, Df = 2, p = 0.050), with the primed-evolved pathogen exhibiting the highest priming sensitivity compared to the ancestral pathogen (p = 0.058). **B** Fitness in the host of each bacterial line was calculated as *mortality proportion X spore load.* Selection treatment and host environment significantly influenced host fitness ($X^2 = 40.385$, df = 5, p < 0.001). Pairwise comparisons confirmed that all the selection treatment levels have lower fitness in the priming host environment. **C** Pearson's correlation between priming sensitivity and average fitness in the host (mean of fitness in primed and control host environment). A, C, and P stand for evolution treatment levels: Ancestral, Control, and Primed evolved pathogen, respectively.

differences, we first analyzed the data by pooling all evolved lines together. Building on this, we then performed statistical comparisons for each experimental line relative to the ancestral strain, allowing us to pinpoint specific evolutionary changes.

Overall, the expression in the evolved lines did not differ significantly from that in the ancestral strain (C: estimate = -0.7565, Df = 14, t = -0.597, p = 0.560; P: estimate = -1.1444, Df = 14, t = -0.903, p = 0.382). Four of the control- and three of the primed-evolved lines showed significantly reduced *cry3a* gene expression compared to the ancestral strain (S3 Table and S3 Fig). However, we found no association between *cry3Aa* expression and virulence of the lines, neither in primed (t = 0.279, Df = 15, p = 0.783) nor in control host environment (t = 0.131, Df = 15, p = 0.897)

## Do the evolved pathogens differ in their genomes and mobilome?

To investigate whether the observed phenotypes of the evolved *Btt* lines are based on genetic changes, we performed whole genome sequencing of all evolved lines and three replicates of the ancestral pathogen. Compared to the ancestral

pathogen, the evolved lines showed a low number of single nucleotide variants (SNVs) and structural variants (SV). Overall, 32 unique variants were identified. The primed-evolved lines contained 30 variants and the control-evolved lines contained 28 variants. Only a few of those variants were shared across different replicates and no distinctive pattern could be observed between the primed and control evolution regime (Fig 5). Common variants in at least 50% of lines included a G to T substitution in an intergenic region between a hypothetical protein and a putative NTP pyrophosphohydrolase (g.2017894G>T), and a small insertion within a hypothetical protein coding sequence (g.2776026insA), causing a frame-shift (Fig 5). Lines C1 and P8 both had unique variants in the same hypothetical protein DUF4430 at different positions: while C1 had a small deletion (g.3358458delC), P8 had an insertion (g.3357970insCAGCTGTTCGTTATATAAATACG-CAA). Both variants caused frameshifts, with the P8 insertion also introducing a premature stop codon.

As most bacteria are host to a multitude of mobile genetic elements that can influence the performance in changing environments, we aimed at identifying differences in such elements between the evolution regimes. The NCBI reference sequence [48] contains six plasmid contigs where the two biggest plasmids are 250kb and 188kb, followed by two plasmids of 77kb, 68kb and the smallest 15kb plasmids. The 44kb plasmid is a phagemid, a circularized phage, which additionally is located on the chromosome of the reference sequence and thus was not included in the plasmid coverage analysis. The analysis for plasmid coverage in comparison to chromosome coverage indicated a decrease in the abundance of the 188kb plasmid in the evolved lines compared to the ancestral lines (Fig 6A). However, only line P8 retained

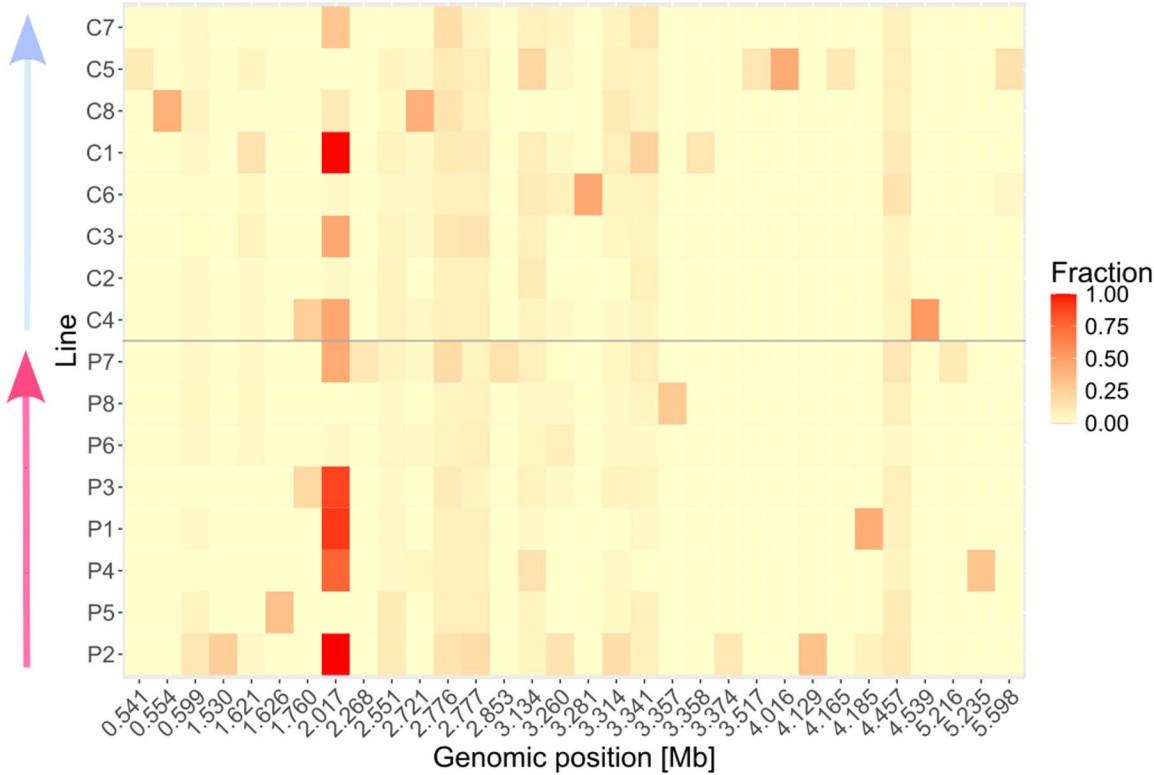

**Fig 5. Heatmap displaying the identified single nucleotide variants (SNVs) and structural variants (SVs) at the respective genomic positions.** Variants in any line with an alternative allele fraction of >=10% of total read coverage are displayed. Alternative allele fraction of each variant for all remaining lines was calculated by dividing alternative allele coverage by total read coverage at that position (red = 100% alternative allele or fraction 1, yellow = 0% alternative allele or fraction 0). Individual lines are ordered within the respective evolution treatment by virulence (cf. Fig 2), from low to high (arrows).

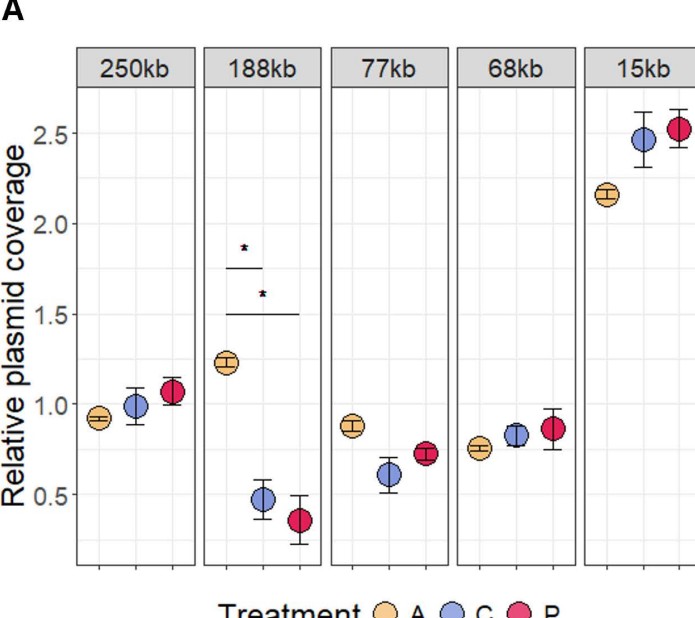

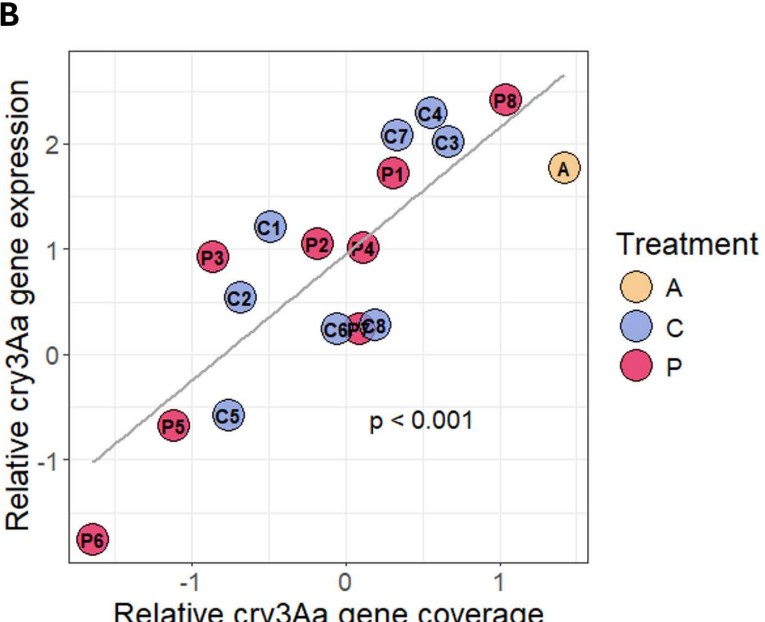

**Fig 6. A Log2 transformed plasmid coverage divided by chromosome coverage (relative plasmid coverage).** Shown are the respective mean values and their standard error. * adjusted p<0.05 **B** Correlation of the Log2 transformed cry3Aa gene coverage divided by chromosome coverage (relative cry3Aa gene coverage) and delta CT of housekeeping gene expression subtracted by cry3Aa gene expression (relative cry3Aa gene expression). The p-value from the linear regression analysis (lm(Cry_expr~Cry_cov)) indicates a significant effect of cry3Aa gene coverage on cry3Aa gene expression.

a plasmid-to-host ratio similar to the ancestral for this plasmid (S3 Fig). This particular plasmid encodes for the important crystal toxin Cry3Aa and other virulence factors (Cry15Aa, Sphingomyelinase C). Additionally, we observed correlation of the log2 transformed Cry3Aa gene-to-chromosome ratio and the relative expression of the *cry3Aa* gene measured via

qPCR expression previously described (Figs 6B and S3). However, correlation of other phenotypic traits with active mobile elements showed no statistical significance (S7 Fig). The coverage plots across the genome contained peaks in distinct areas of the chromosome (S4 Fig) that partially overlapped with annotated prophage areas known as *Bacillus* phages [49]. We found that the phages phBC6A52, phi4J1 and phiCM3 had the greatest differences in phage-to-host ratio (Fig 7). We performed double-layer agar assays (DLAs) for two out of 8 lines of each evolution environment and the ancestral strain to verify that phages were active in the evolved lines, but not in the ancestral strain. As expected, we observed visible round plaques in the bacterial lawns of the evolved lines, but not the ancestral lines (S6 Fig).

## Discussion

To investigate the potential effects of immune priming on virulence evolution, we conducted one-sided experimental evolution with *Btt*, using either primed or non-primed control *T. castaneum* beetle hosts, and evaluated the phenotype and genotype of the evolved pathogen lines. While average virulence (measured as the proportion of dead hosts), did not differ strongly between these selection environments, we identified priming as a selective pressure leading to increased variation in virulence when tested in the control host environment (Fig 2D). Some primed-evolved lines showed higher virulence than any of the control-evolved lines and similar to the ancestral *Btt*, whereas other primed-evolved lines caused the lowest host mortality rates (Fig 2A). Another important observation is that the primed-evolved pathogens did not adapt better to the primed host environment. Host priming remained effective and reduced mortality also against the primed-evolved pathogens (Fig 2A), which were moreover not able to increase spore production in primed hosts (Fig 3B). This implies that the primed-evolved pathogens failed to develop mechanisms to evade the host's priming immune response. Even the most virulent lines did not overcome the primed immune response, indicating that priming serves as a robust protective measure for the host.

Although in both our experimental evolution regimes we extracted bacteria from larvae that died by infection – selecting for high virulence, we observed lower virulence in the evolved lines than in the ancestral *Btt.* This can probably be attributed to one-sided adaptation in *Btt*, as the host lacks the opportunity to evolve. Previous studies have shown that high virulence may not be favoured due to its associated costs when host co-adaptation is restricted [10,50]. Additionally, because certain virulence factors are costly and encoded on mobile elements, bacteria may lose them, particularly when propagated in a medium, as demonstrated in previous research [10,47]. Indeed, we found a reduction in the overall coverage of the 188kb plasmid encoding for several virulence factors, as well as differences in the expression of the *cry3Aa* gene that correlated with its genomic coverage (Fig 7). A reduction in Cry toxin production and plasmid carriage during passaging experiments has been shown in other studies using *B. thuringiensis* [51]. Such losses can be explained by the advantage of "cheating" cells within a population of *Bt* that do not express *cry3Aa* anymore and thereby gain a growth advantage [51]. However, in our experiment, *cry3Aa* expression alone did not explain variation in virulence, and the replicate with the low virulence (P2) did not change in *cry3Aa* expression compared to the ancestral strain, whereas the replicate with the high virulence (P7) showed significantly lower *cry3Aa* expression (S2 Fig). Although surprising, this result might be related to the less straightforward role of the Cry3Aa toxin for *Btt* virulence in *T. castaneum* [51,52], compared to other insects. For example, in contrast to other systems, purified Cry toxin alone does not induce beetle mortality, but rather a combination of spores and toxins is required for lethal effects [46,52]. Moreover, a recent study indicates that Cry toxins are involved in the induction of the priming response [53]. Thus, evolution in primed hosts might select against high Cry expression, which in turn could make other virulence factors such as chitinases, metalloproteases or vegetative insecticidal proteins more relevant in these evolved lines [41,54]. Finally, previous studies have shown that evolution can result in reduced virulence due to the interaction with defensive microbes of the host [19] or due to antagonistic interactions among different pathogen strains [55]. A previous study has shown that immune priming leads to changes in the microbiome, specifically by increasing the abundance of pre-existing *Bacillus* taxa [56]. This shift in microbiome composition could create a competitive environment in the beetle gut, where the infecting *Btt* pathogen experiences antagonistic interactions with resident *Bacillus* species.

**A**

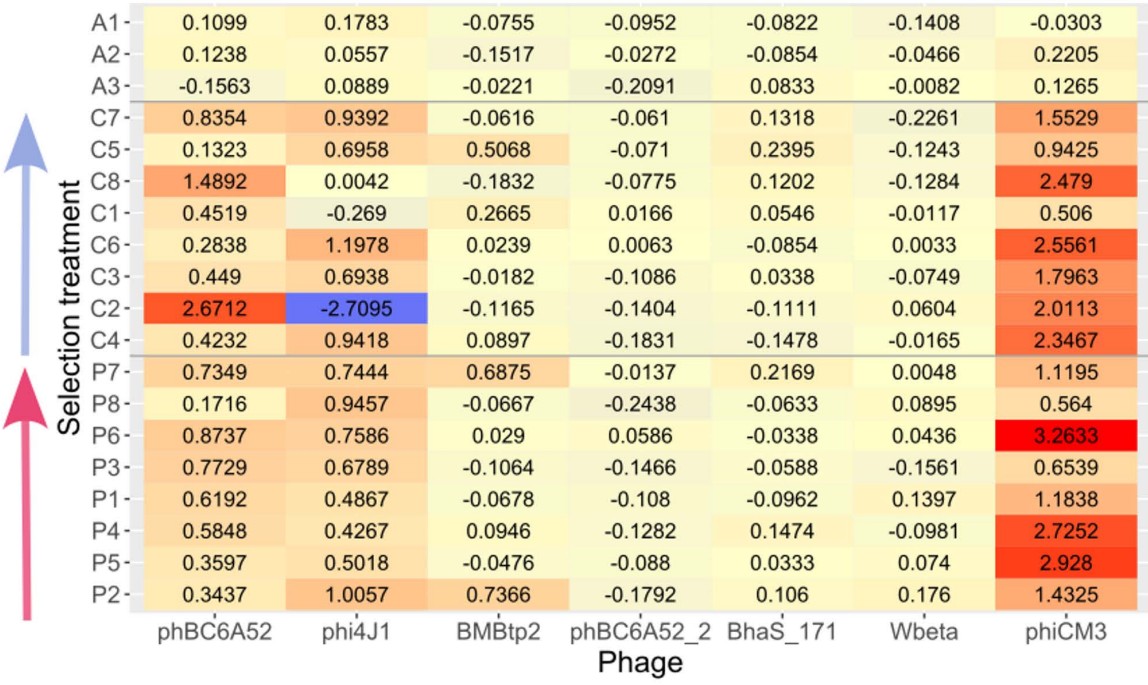

**B**

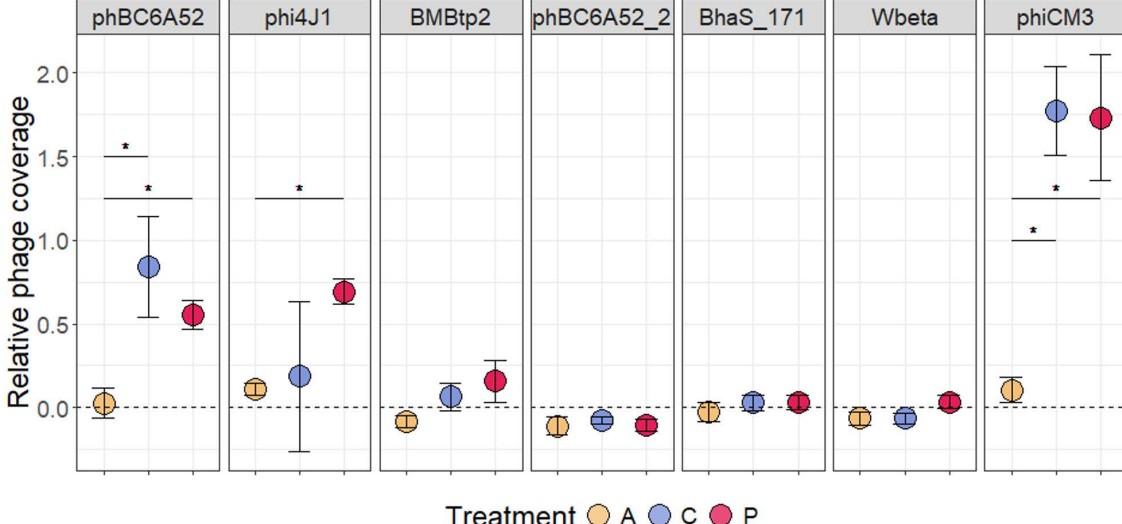

**Fig 7. Analysis of log2 transformed values of phage coverage divided by chromosome coverage (relative phage coverage). A =** Heatmap for each replicate line. A = ancestral, C = control evolved, P = primed evolved. The lines are ordered by evolution treatment and by virulence where the arrow indicates the increase of virulence. Colour gradient indicates high to middle to low relative phage coverage (red to yellow to blue). **B** Mean values of relative phage coverage for selection treatment and their standard error. * adjusted p < 0.05.

Compared to ancestral *Btt*, evolved *Btt* showed reduced sporulation in liquid medium as well as in primed hosts (Fig 3). This decline in sporulation may be attributed to the presence of active prophages that we identified in the genomes of the evolved lines (Fig 7). The activation of prophages and the transition from lysogeny (dormant state) to the lytic cycle (active viral replication and bacterial cell lysis) are often triggered by environmental factors [57]. Such stress can occur under constant sporulation conditions and is known to cause losses in other *Bacillus* fermentation and to interfere with the evolution of virulence of phages towards their bacterial host [58,59]. Interestingly, that a reduction of spore production of evolved lines occurred only in primed but not control hosts (Fig 4A and 4B) could suggest that phage induction is primarily promoted in primed hosts. This environment represents a more stressful niche for the pathogen, as priming leads to the damage in the gut of *T. castaneum* larvae, triggering an early physiological stress response along with the upregulation of immune related genes [60]. Additionally, antagonistic competition may maintain these phages [61]. *Bacillus* is a member of the microbiome of *T. castaneum* and immune priming can cause a shift towards higher abundance of *Bacillus* species [56]. Releasing lytic phages could provide a competitive advantage by targeting members of the same genus or species.

While observing changes in phages and plasmids, we could not pinpoint any common chromosomal mutations that are specific to the selection treatments (Fig 6). This does not exclude that there might be more subtle genetic changes that went undetected because they might be specific to certain replicates. Indeed, the observed increase in variation in virulence indicates that replicates may take different evolutionary routes [62] making it difficult to find common genetic changes.

Moreover, the lack of a link between spore load and virulence might suggest that *Btt* virulence and transmission evolve independently. A comparable result was noted in the pathogen *Pseudomonas aeruginosa* following evolution in hosts with partial immunity [17]. Unlike our findings, the incomplete immunity in that study resulted in increased pathogen virulence. Nevertheless, our results are in line with the mentioned study that revealed that the most virulent pathogens did not develop mechanisms to counteract the protective benefits of priming exposure. It is worth noting, however, that in this case, the priming or partial host immunity was non-specific and triggered by the host's microbiota, while in our study priming is specific for the bacterial strain and triggered by pathogen-derived cues present in the supernatants used for priming [39,40,53,60]. Finally, different host-pathogen systems may show divergent evolutionary trajectories.

The increased variance in virulence in primed-evolved hosts might be explained by the more variable niche that a primed host represents for the pathogen [63]. *Btt* in primed *T. castaneum* hosts are confronted with an increased intensity and diversity of immune defences [44], which are characterised by dynamic changes over time, as well as variability among host individuals and lines [28]. Therefore, the hostile host environment of an immune primed host likely represents a more stressful environment for the pathogen. Moreover, pathogens in primed hosts are exposed to a changed microbiome composition [56]. Immune-primed hosts could thus represent a fluctuating environment, which is known to increase phenotypic and genetic diversity [64–66], allowing pathogens to retain the flexibility to adapt to diverse host conditions over time. We indeed found the magnitude of variation in virulence to be much higher in primed hosts than in control hosts (Fig 2E). Moreover, variance in the pathogen population could also result from antagonistic pleiotropy involving active phages that could be beneficial in one, but detrimental in another environment [66]. Lytic phages are well known to be activated in stressful environments and could thereby contribute to variation among pathogens [67].

Interestingly, despite the increased variance, pathogen lines evolved in primed hosts exhibited greater priming sensitivity, showing lower virulence (Fig 4A) in primed hosts compared to ancestral and control-evolved *Btt* lines. This could suggest that the primed environment does not necessarily select for higher virulence in the short term but instead leads to divergent evolutionary trajectories. It is possible that given more generations, clearer patterns of adaptation— increases in virulence—might emerge as our results indicate that the primed-evolved pathogen is 85.15% more likely to have higher virulence than the control The observed trend of a negative association between the fitness of the primed-evolved pathogen and its priming sensitivity suggests that pathogen strains with higher fitness are less affected by the host's primed immune response (Fig 4). This indicates that fitter pathogens may have developed strategies to

more effectively resist or evade the immunological effects of host priming. Previous research has demonstrated that *B. thuringiensis* can suppress the humoral immune system of the diamondback moth [68]. Additionally, it has been found that *T. castaneum* shows a unique immune response when orally infected with *B. thuringiensis*. In particular, genes linked to the Toll and Imd pathways were found to be downregulated after oral infection [69]. Future studies could focus on relevant immune markers in the host, to determine whether evolved pathogens actively suppress immune responses. This would help clarify whether the observed fitness differences among strains result from enhanced immune evasion mechanisms.

Our observation of increased virulence variation may also result from differences in our system compared to models of leaky vaccines, where the pathogen is directly transmitted between vaccinated hosts [12,14,70]. In contrast, *B. thuringiensis*, both in our experiment as well as in nature, experiences two distinct environments, as it can thrive outside of its insect hosts [71]. In addition to host factors, the longevity of spores and their ability to persist in the environment may also play a role in determining virulence [72].

In conclusion, our results provide, to our knowledge. the first evidence of how bacterial pathogens evolve in response to specific immune priming of their hosts. The observed increase in virulence is evolutionarily significant, as variation is essential for selection to act upon. Increased variance in a subpopulation evolving with primed hosts may fuel the evolutionary flexibility of the entire pathogen population, as diversity within bacterial communities significantly influences their survival within hosts [73] enhancing the overall resilience of the population [74,75]. Therefore, host immune priming and other forms of incomplete, innate immune memory, may boost the evolutionary potential of pathogens and even lead to multiple evolutionary stable states [11]. This is important because innate immune memory bears a large potential for applications. The growing importance of trained immunity in medicine is being paralleled by the increasing applications of immune priming in invertebrates, particularly in sectors such as aquaculture and large-scale insect production. Both areas are experiencing significant advancements and garnering increased attention [76,77]. Studies in further systems with immune priming and trained immunity are needed to better understand and potentially manage the evolution of pathogens when confronted with innate immune memory.

## Materials and methods

### Model organisms

We utilized *Bacillus thuringiensis* subspecies *morrisoni* biovar *tenebrionis* (*Btt)* as the pathogen, sourced from the Bacillus Genetic Stock Centre (BGSCID) at Ohio State University, USA, identified by the code 4AA1. The strain was preserved in 500 µL microtubes with 25% glycerol at a temperature of -80°C. This strain has been shown to cause mortality in *T. castaneum* larvae upon the oral ingestion of spores and their toxins [42,43].

As the host, we used a wild-type *T. castaneum* (Cro1) population collected from Croatia in 2010 [42] and adapted to laboratory conditions for eight years in numerous non-overlapping generations until the start of the experiment in 2018. Beetles were reared on organic wheat flour (Bio Weizenmehl Type 550, DM-drogerie Markt GmbH + Co. KG) supplemented with 5% brewer's yeast (the flour mixture was heat-sterilised at 75°C for 24 h). For rearing and infection experiments, beetles were kept at 30°C, 70% relative humidity, and a 12-h/12-h-light–dark cycle.

### Experimental evolution

**Preparation of bacteria for selection treatment and infections.** We prepared spore cultures and supernatants as previously [39,40], with minor modifications. To begin the selection experiment, we plated *Btt* vegetative cells from the -80°C glycerol stock on an LB agar plate and incubated it overnight at 30°C. The following day, we inoculated 5 ml of Bt medium (w/V–0.75% Bacto Peptone (Sigma), 0.1% glucose, 0.34% $KH_2PO_4$, 0.435% $K_2HPO_4$) that was supplemented with 25 µl of a sterile salt solution (0.2 M $MgSO_4$, two mM $MnSO_4$, 17 mM $ZnSO_4$, and 26 mM $FeSO_4$) and 6.25 µl of

sterile 1 M CaCl$_2$×2H$_2$O with a single colony, and incubated it at 30°C and 200 rpm overnight. Overnight cultures were moved to 1-L Erlenmeyer flasks containing 300ml of Bt medium supplemented with 1.5mL of salt solution and 375 μL 1 M CaCl$_2$×2H$_2$O. Since *Btt* is naturally neomycin-resistant, 15μg/μL of neomycin were added to the sporulation media. This step was taken to guarantee that other spore-forming bacteria that could come from cadavers did not proliferate in the medium. The cultures were kept at 30°C and rotated at 180 rpm for seven days in the dark. On the third day of incubation, 1.5mL of salt solution and 375 μL of 1 M CaCl$_2$×2H$_2$O were added to the cultures. Following seven days of sporulation, the culture was spun for 20 minutes at 4500 rpm. The pellet was washed and redispersed in PBS (Calbiochem). Using means of flow cytometry (FACSCanto II; Becton Dickinson, USA) as previously described [43] spore concentration was adjusted to 5 x 10^9/mL, and the diet was prepared by adding heat-sterilized flour with yeast (0.15 g/mL of spore suspension; [42,43]. We then pipetted the liquid spore-containing diet (10 μL per well) into the 96 flat bottom microtiter plates (Sarstedt, Germany) under sterile conditions, covered them with a breathable sealing foil for culture plates (Kisker Biotech), and dried them overnight (~17 hours) at 30°C. For all remaining selection cycle experiments, where spores were taken from cadavers, we slightly adjusted the protocol; after processing spores, we inoculated 50 mL of Bt medium supplemented with salts and neomycin in a 100 ml Erlenmeyer flask, followed by seven days incubation at 180 rpm and 30°C.

**Preparation of primed and control host for the selection treatment.** The primed host environments were performed as previously described [39,40]. *Btt* vegetative cells were plated from a -80°C glycerol stock on LB agar plates and incubated overnight at 30°C. Five single colonies were incubated overnight, and sporulation cultures were produced as described above but without adding neomycin. After seven days of sporulation and centrifugation, the collected supernatants were centrifuged twice at 4,500 rpm for 15min and then filter-sterilized with a 0.45-μm pore-size, followed by a 0.2-μm pore-size cellulose acetate filter (Whatman GmbH). We mixed every millilitre of spore-free supernatant with 0.15 g heat-treated flour (supplemented with 5% yeast). Next, ten microliter of the priming diet mixture was pipetted into each well of a 96-well plate (Sarstedt, Germany), sealed with a breathable foil (Kisker Biotech), and dried overnight at 30°C. For the control treatment, the diet was prepared by mixing 0.15 g heat-treated, yeast-supplemented flour with each millilitre of sterile non-conditioned growth medium.

For every selection treatment cycle, we took approximately 2,000 1-month-old adults and left them to lay eggs for 24 hours on the flour. The priming protocol was as follows: 15 days after oviposition, 192 larvae per replicate line were individually placed on a previously prepared priming diet in two 96-well flat-bottom plates. The plates were sealed with clear tape and punctured with needles to allow air circulation and were stored under standard rearing conditions. After 24h on a priming diet, we transferred larvae to flour discs consisting of phosphate-buffered saline (PBS) mixed with 0.15g/mL flour, in which they stayed for four days until we transferred them again to pathogen spore-containing discs.

**Re-infection for another selection treatment cycle.** We measured mortality for seven days and then collected larvae that died during the first two days, placed them in the dark, and waited for another seven days to ensure that all *Btt* cells in the cadaver had sporulated. For reinfection, we isolated spores from insect cadavers and propagated them once in liquid sporulation media to obtain a sufficient number of spores for infection. For each bacterial line, we isolated spores from five cadavers, which were subsequently grown independently in a growth medium to achieve high spore numbers for the infection of a sufficient number of beetle larvae for the next infection cycle. Spores from the five cultures derived from the cadavers were combined for infection to reduce the effect of potential clonal interference among rapidly growing clones in culture. We randomly selected five cadavers per individual line and individually homogenized them as described before [43]. We then counted spore concentrations using flow cytometry and pasteurized the suspensions at 80°C for 10 minutes (i.e., any possible vegetative cells are killed). Liquid medium was inoculated with 1mL of 1x10^6 spores derived from cadavers for germination, followed by spore production. After seven days of incubation, we pooled all five individually raised spore cultures and prepared the spore-flour diet. Primed and control host larvae were then placed on spore flour discs, and their survival was tracked.

### Phenotypic readouts

**Virulence and spore production of evolved bacterial lines.** All the selection treatment lines were introduced to a primed and control host environment. Host mortality was screened for six days. We performed this experiment in three experimental blocks, with 48 larvae per treatment/block, resulting in 144 larvae per treatment combination. For every block, we took approximately 2,000 1-month-old adults and left them to lay eggs on the flour (implemented with 5% yeast) for 24 hours, following the priming protocol described above. For each block, we prepared 12 96-well plates with a priming diet and 12 96-well plates with control diet. We placed 15-day-old larvae on the plates and, after 24 hours, transferred them to PBS flour discs. After four days, we transferred them to 96-well plates containing spores from 24 bacterial lines (eight Primed evolved, Control evolved and Ancestral). Each 96-well plate contained spores from all 24 bacterial lines to control for the possible plate effect. This resulted in 24 96-well plates for each experimental block.

Spores from the bacterial lines were prepared as described above. Frozen spores from the 8th passage and ancestral *Btt* were first plated on an agar plate and after 14 hours 10 colonies were inoculated in 50 mL growth medium supplemented with salts. After seven days of bacterial growth and sporulation, spores were washed, and measured via flow cytometry, and the concentration was adjusted to $5x10^9$cells/mL. We used measures of the numbers of spores in a liquid medium from three experimental blocks to test for the differences in spore production between the lines.

We quantified the pathogen load of infected larvae that died from the infection. For this, we took ten cadavers per bacterial line, which resulted in a total of 240 cadaver samples. The samples were processed as described before [43]. Seven days after their death, each cadaver was placed in a 100 µl Eppendorf tube with 50 µL PBS and a sterile metal bead. The cadavers were homogenized using a Mixer Mill MM301 (Retsch) to release spores into the solution. The solution was then washed using a cell strainer to remove larger particles. The solution was diluted 1:1000 and measured via flow cytometry.

We quantified *Btt* fitness by calculating the product of spore load and host mortality rate. As an obligate killer, *Btt*'s reproductive success relies on both its capacity to infect and kill hosts and the number of spores produced per infected host.

**Cry gene expression in Btt cultures.** Before RNA extraction, we inoculated two cultures of ancestral and evolved lines in 6 mL LB medium at 30°C, 180 rpm. Following 12 h of incubation, the cultures were centrifuged at 4500 rpm for 10 min and resuspended in 100 µL of PBS. We added 2.2 µl of lysozyme (100 mg/mL) to 100 µL of the bacterial suspension and incubated for 30 min at 37°C with shaking. Subsequently, we implemented an RNA extraction protocol with a combination of TRIzol reagent (Ambion, USA) lysis and extraction with the purification via spin columns from the SV Total RNA Isolation System (Promega) as described before [78]. To analyze *cry3a* expression, cDNA synthesis, and qPCR were performed in two identically designed blocks. We conducted reverse transcription of cDNA using the RevertAid First Strand cDNA Synthesis Kit (ThermoScientific) following the manufacturer's instructions. For the reaction, we used 2µL of extracted RNA, irrespective of its concentration, along with a random hexamer primer. The resulting cDNA product was then diluted 1:5 with nuclease-free water.

Following cDNA synthesis, we performed qPCR. This analysis involved measuring the expression of *cry3a* relative to two housekeeping genes (*yqey* and *rps21*) (S1 Table). Prior to the main experiment, we determined the amplification efficiency of the primers by qPCR using a serial 10-fold dilution of the initial cDNA input (S1 Table).

### Genome analysis

**Sequencing.** After culturing samples from all evolved and three ancestral replicate lines on LB agar plates, we extracted DNA using the Monarch Genomic DNA Purification Kit (New England Biolabs GmbH, Frankfurt am Main, Germany). For each evolutionary line, we inoculated 10 independent overnight cultures from single colonies, extracted DNA from these cultures, and pooled the samples before sequencing. This approach allowed us to capture the genetic diversity within each evolved population rather than focusing on individual clones. For sequencing on the PacBio Sequel IIe platform (Pacific Biosciences Inc., Menlo Park, CA, USA), we created a library with the SMRTbell Express Template

Prep Kit 2.0 (Pacific Biosciences Inc.) according to the manufacturer's recommendations, aiming for at least 20x coverage on the HiFi read level. We then mapped the resulting HiFi reads against the sequence of a *Btt* reference strain (NCBI accession no.: GCF_022810725.1) using the SMRT Link software version 11.

**Variant detection.** To call single nucleotide variants (SNVs) we used GATK Haplotypecaller (v4.3.0.0) [79] and Google's DeepVariant [80] on the Galaxy Server (Galaxy Version 1.5.0 + galaxy1). We then intersected the resulting vcf files of the evolved lines with those derived from the ancestral lines using the bcftools isec function (version 1.16) [81] to filter out variants that were based on differences between our ancestral lines and the used reference sequence. Of the remaining SNVs we kept those with an alternative allele fraction of >= 10% of the total read coverage at the respective position. To call structural variants (SVs) we used Sniffles2 (Version 2.2) [82], The resulting vcf files were intersected in the same way as the SNV vcf files. Next, we verified all final variants by manually inspecting them in the Integrative Genomics Viewer (IGV) [83]. To annotate the SNVs we used SnpEff (version 5.2c (build 2024-04-09 12:24)) [84] using the prokaryotic codon table and the reference genbank- and nucleotide fasta files.

**Plasmid coverage.** We calculated the coverage across the chromosome and the different reference plasmids respectively using samtools depth (version 1.16.1) [81] and estimated the abundance of plasmids by dividing each plasmids mean coverage by the chromosomes mean coverage and taking the log2 of those values. We termed the resulting values "Log2(Plasmid-to-Host ratio)".

**Phage analysis.** To detect potentially complete prophage genomes within our reference genome we used PHASTER [85]. Next, we calculated the coverage across the genome using samtools depth and divided the genome into sections of a) outside of prophage regions and b) within prophage regions. We considered b) to be true if the region contained all essential phage genes necessary for its reproduction (from small terminase to lysis cassette). Finally, we divided the means of b) by the means of a) and took the log2 of those ratios. We termed the resulting values "Log2(Phage-to-Host ratio)". The log2 transformed Phage-to-Host ratios were then plotted using ggplot2 in Rstudio. To confirm the presence of active phages in the evolved lines, we performed double-layer agar assays (DLA). In short, we grew bacteria in 20 ml Luria broth (LB) medium overnight. The next morning, we adjusted the overnight culture to an optical density of 1 at 600 nm wavelength ($OD_{600}$ of 1) and inoculated 500 µL into 50 ml fresh LB medium. After four hours, 8 hours, and 12 hours, we took 1 ml of the culture and adjusted it to an $OD_{600}$ of 1.5. We then mixed 400 µl of the culture with 4 ml lukewarm 0.7% LB agar and poured the mix onto a solid 1.5% LB agar plate. After 24 h, we checked the plates for plaques in the bacterial lawn.

## Statistics

All data were analysed using R [86] and R Studio [87]. To test whether the pathogen selection treatment and host environment have an effect on the virulence, we tested the effect of selection treatment (primed-evolved, control-evolved and ancestral) on the host survival by fitting a generalized linear mixed effect model with interaction (GLMM) and binomial error distribution using *the glmer* function in the package *lme4* [88] (mortality~ selection_treatment*host environment) +(1|Replicate_line)+ (1|Block)+(1|Plate). The selection treatment level and host environment were defined as fixed factors, whereas a bacterial line, plate and block were defined as random factor. As interaction was not significant, we then fitted a model with selection treatment as a fixed factor in primed and control host environments separately. To further test the selection influence of mortality variation, we ran a Bayesian Beta regression model in rstan [89], allowing the precision parameter (φ) to be influenced by the Selection treatment. Traditional models often assume constant variance in survival models, which may not hold when survival rates vary across groups. The Beta distribution allows flexible modeling of survival probabilities, capturing different survival patterns while accounting for heteroscedasticity. Its variance depends on both the mean survival proportion (μ) and the precision parameter (φ), ensuring a more accurate representation of uncertainty in survival outcomes. We used the model's posterior samples to estimate the probability that mortality variation arising from primed, control, and ancestral populations are different. The variance of the response variable in a beta

regression depends on both μ and ϕ, calculated as Var(Y)=μ(1−μ)/ 1+ϕ. We used this estimated variance to estimate the K-CV, a more robust measurement of covariation [90]. We used the posterior estimates of K-CV and Var(Y) to calculate the fold change (log2) of virulence variation for the control and prime pathogen lines and report the mean and 95% Highest Density Interval.

For spore load (transmission) analysis, the *lmer* function in the *lme4* package was used with the bacterial line as a random factor (spore_load~ selection_treatment +(1|replicate_line) in primed and control host environment separately. For spore growth analysis we performed linear regression analysis with *lm* function. After fitting the models, the variance between the selection treatment (primed-evolved, control-evolved, ancestral) was assessed using one-way analysis of variance. The means were compared using Tukey's post-hoc analysis with Benjamini-Hochberg correction.

Priming sensitivity of each bacterial replicate was determined by dividing proportion of mortality in control host with proportion of mortality in primed host. Kruskal-Wallis test was used to assess differences in priming sensitivity between the selection treatment levels. The means were compared using post-hoc pairwise comparisons using Dunn's test with a Benjamini-Hochberg correction.

We determined fitness in the host by combining spore load and proportion of mortality per individual bacterial line. Then we compared it between selection treatment levels using *lmer* function (Fitness_in_host~Line * Environment + (1 | Replicate_line). The means were compared using Tukey's post-hoc analysis with Benjamini-Hochberg correction.

To analyze gene expression data from qPCRs, we first compared the relative expression of *cry3a* of the primed and control-evolved *Btt* to the expression in the ancestral *Btt* cultures in a linear mixed model with experimental block and evolved replicates as random factors (relative expression~selection_treatment + (1|block) + (1|replicate)) using the packages *lme4* and lmerTests [91]. We also fitted a second linear mixed effects model using the same packages to compare the evolved lines to the ancestral cultures (relative expression~selection_treatment + (1|block)).

We used various statistical techniques to evaluate the relationships among different variables. The connections between spore load and mortality, mortality and cry expression, and average host fitness with priming sensitivity were analyzed using Pearson's correlation method. We conducted Mann-Whitney tests using the wilcox.test() function to evaluate the significance of differences between log2 transformed Plasmid-to-Host and Phage-to-Host ratios. The p.adjust() function with the "holm" adjustment method was applied to address multiple testing concerns. A linear model was constructed to investigate significant interactions between the read coverage for the cry3Aa gene and its expression. The cor() function from the R Stats package (version 4.3.1) was used to determine the correlation strength between different phenotypic readouts and active mobile elements for both primed evolved lines and control evolved lines. To statistically compare the correlation strengths of various readouts between primed and control evolved lines, we utilized the cocor() package (version 1.1-4) [92].

## Supporting information

**S1 Table. qPCR primers and their amplification efficiencies.**
(DOCX)

**S2 Table. Relative expression of Cry3a gene in cultures from different Btt strains.** Btt- and Bt407- do not carry the Cry3a gene. Priming and control lines were evolved from ancestral strain. Mean ΔCt values by subtracting the target gene Ct value from the geometric mean of the Ct values of the housekeeping genes.
(DOCX)

**S3 Table. Differences of Cry3a gene expression of individual evolved lines compared to the ancestral strain.** Shown are the results of linear mixed effects model with block as experimental factor. Significant differences are shown in bold type face.

(DOCX)

**S1 Fig. Correlation between spore load and virulence in ancestral line (A), control line(C) and primed line (P) in primed hosts.**
(DOCX)

**S2 Fig. Correlation between spore load and virulence in ancestral line (A), control line(C) and primed line (P) in primed beetle larvae in control host.**
(DOCX)

**S3 Fig. Relative expression of Cry3a gene for evolved lines.** ΔCt values were calculated by subtracting the Ct value of the target gene, Cry3A from the geometric mean of the Ct values of two housekeeping genes (Yqey and rps21).
(DOCX)

**S4 Fig. Heatmap for the Log2 transformed values of plasmid coverage divided by chromosome coverage for each replicate line.** A=ancestral, C=control evolved, P=primed evolved.
(DOCX)

**S5 Fig. Exemplary log2 transformed mean-normalized coverage plots of reads across the chromosome for control (C1 – 4) and priming (P1 – 4) evolved lines.**
(DOCX)

**S6 Fig. Double layer agar assays of an ancestral line A and evolved lines C1 (control evolved) and P7 (priming evolved).**
(DOCX)

**S7 Fig. Pearson correlation matrix for the primed evolved pathogen (A, C) and the control evolved pathogen (B, D).** Correlated traits include all phenotypic traits with active mobile elements (A, B) or Fitness and Priming sensitivity values with active mobile elements (C, D).
(DOCX)

## Acknowledgments

We thank Nicholas Carl Heinrich Schröder, Katharina Natascha Meyer zu Riemsloh and Kathrin Brüggemann for assisting in the lab during experimental evolution, Stefan Bletz for supporting whole genome sequencing procedures, Barbara Milutinović for valuable comments and suggestions, and Helle Jensen for the drawings of larvae.

## Author contributions

**Conceptualization:** Ana Korša, Joachim Kurtz.

**Data curation:** Ana Korša, Moritz Baur, Nora K. E. Schulz, Alexander Mellmann.

**Formal analysis:** Ana Korša, Moritz Baur, Nora K. E. Schulz, Jaime M. Anaya-Rojas.

**Funding acquisition:** Joachim Kurtz.

**Investigation:** Ana Korša, Moritz Baur, Nora K. E. Schulz, Alexander Mellmann.

**Supervision:** Joachim Kurtz.

**Visualization:** Ana Korša, Moritz Baur, Jaime M. Anaya-Rojas.

**Writing – original draft:** Ana Korša, Moritz Baur, Joachim Kurtz.

**Writing – review & editing:** Ana Korša, Moritz Baur, Nora K. E. Schulz, Jaime M. Anaya-Rojas, Alexander Mellmann, Joachim Kurtz.

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
