## [Decision Letter · Decision Letter 0]

4 Mar 2025

PPATHOGENS-D-24-02760

Experimental evolution of a pathogen confronted with innate immune memory increases variation in virulence

PLOS Pathogens

Dear Dr. Korša,

Thank you for submitting your manuscript to PLOS Pathogens. After careful consideration, we feel that it has merit but does not fully meet PLOS Pathogens's publication criteria as it currently stands. Therefore, we invite you to submit a revised version of the manuscript that addresses the points raised during the review process.

Please submit your revised manuscript within 60 days May 03 2025 11:59PM. If you will need more time than this to complete your revisions, please reply to this message or contact the journal office at plospathogens@plos.org. Please include the following items when submitting your revised manuscript:

We look forward to receiving your revised manuscript.

Kind regards,

Robert L. Unckless, Ph.D.

Academic Editor

PLOS Pathogens

Debra Bessen

Section Editor

PLOS Pathogens

 Sumita Bhaduri-McIntosh

Editor-in-Chief

PLOS Pathogens

orcid.org/0000-0003-2946-9497

 Michael Malim

Editor-in-Chief

PLOS Pathogens

orcid.org/0000-0002-7699-2064

**Additional Editor Comments (if provided):**

Three experts in the field have reviewed your manuscript entitled "Experimental evolution of a pathogen confronted with innate immune memory increases variation in virulence". All three were enthusiastic about the work but pointed out significant concerns. All three reviewers point to references in the literature that should be considered in a revision and to a more careful grounding of the current hypotheses in terms of the available literature. From an editorial perspective, my biggest concern was raised by Reviewer #2 who asked whether the increased variance observed was the result of comparing technical to biological replicates in the experimental vs. control populations. If so, can we really say anything about increasing variance in the primed populations? All three reviewers also asked for more clarity in discussing aspects of the results (implications for selection, justification for fitness proxies, discussion of other virulence factors). Finally Reviewer #2 suggests a few additional experiments (pathogen load, pupation rate). I wonder whether you have that pupation rate data and could include it as another measure of fitness.

**Journal Requirements:**

**Reviewers' Comments:**

Reviewer's Responses to Questions

**Part I - Summary**

Reviewer #1: This study uses an experimental evolution approach to test whether innate immune priming selects for higher pathogen virulence, using red flour beetles and their Bacillus pathogen as a model system. Understanding effects of innate immunity on virulence evolution is a very important and relevant question in many systems, and the authors use an ideal model system to test this question. Overall the methods and analysis are well done, and the results are quite interesting in that virulence did not increase, but became more variable after experimental evolution in primed hosts. I think overall that this is an exciting and important contribution to the literature.

Reviewer #2: The authors examined how immune priming by pathogens influences the evolution of virulence. While virulence did not differ across primed and control host treatments, there was greater variation in terms of virulence across replicates in primed hosts. Although this particular result was interesting, I have several concerns:

Line 95: The claim that this is the first study to show the influence of immune priming on pathogen evolution is not accurate. Previous studies, which the authors cited, have examined the interaction between live pathogen exposure (Read 2015), antigen exposure (Barclay 2012), and microbiota exposure (Hoang 2024), on the immune system and how they shape pathogen evolution. As immune priming is a general phenomenon, saying that immune priming has not been examined before in the context of pathogen evolution is misleading.

Line 360: With regards to specificity: has it been shown that nothing else could have triggered the same response from the beetle host?

The significant finding of this study, where variation increased across replicate populations, is interpreted as “evolutionary relevant because selection needs variation to act upon” (line 388). This interpretation is incorrect has evolution acts on the variation within a population, not across independently evolving populations.

Line 164: The increase in variation from the ancestor is due to the ancestor having technical replicates instead of biological replicates? So an increase in variation would be expected.

Lines 366 – 367: the way this is worded suggests that immune priming has been shown to maintain microbial variation through balancing or pleiotropic effects, when neither citation has to do with immune priming.

The Discussion was a bit too long in general.

Reviewer #3: In this study, Korša et al asked whether immune priming in the insect host Tribolium castaneum result in the evolution of increased virulence in the entomopathogenic bacteria Bacillus thuringiensis tenebrionis (Btt). Using a one-sided experimental evolution set-up, they subject Btt to undergo selection pressure in either an immune primed host or non-primed host for 8 selection cycles. To test the effects of their selection treatments, they performed a common garden experiment where evolved-lines of Btt and the ancestral strain were phenotyped for host-mortality (virulence) and spore load (transmission) in either an immune primed host or non-primed host environment. The authors observed that both evolved Btt lines did not increase virulence or transmission, in-fact they were reduced, when infecting primed or non-primed hosts. Interestingly, they found that the selection pressure of immune priming resulted in increased variation in Btt virulence. Genomic sequencing of these evolved strains revealed, when compared to the ancestral strain, minimal chromosomal mutations and reduced covered of the plasmid 188 kb, known to carry genes encoding virulence factors like the Cry toxin. The authors find some evidence that the evolved Btt lines have increased lytic phage activity, which could potentially explain reduced sporulation in those lines. While some additional experiments can help complete the presented work, this is an interesting study that interrogates whether incomplete immunity like immune priming can act as a selection pressure for bacteria pathogens to evolve virulence mechanisms against. The study appears mostly sound; however, the authors should revise the language to improve readability.

**Part II – Major Issues: Key Experiments Required for Acceptance**

Reviewer #1: My major concerns are all regarding presentation / interpretation. The main result is higher variation in virulence in pathogen lines evolved in primed hosts. This is quite interesting and I think can be highlighted even more strongly. I also would suggest using something other than variance as your metric presented in the results-I would encourage the authors to look at this paper which has been very helpful to my own lab group in thinking about how best to quantify differences in variability between groups: https://besjournals.onlinelibrary.wiley.com/doi/full/10.1111/2041-210X.14197

In terms of interpreting the variance result, in the discussion there is very little time spent on the implications of this for selection. At the very end, you note that such variation may boost evolutionary potential (line 389), yet you also saw no signal of directional selection because of this variability, which you note may reflect different evolutionary routes taken by each line. I think these implications are worth more focus in the discussion- do you think that evolutionary potential truly is higher in the primed treatments, and you just needed more generations of selection to see any signals of evolution? Overall I would have liked to see more extended discussion of the implications of what, to me, was a really interesting and exciting result (higher variation in bacterial lines evolved in primed hosts). This seems really novel and exciting, and broadly parallels some results we are finding in bacterial gene expression in songbirds with and without prior pathogen exposure (still as yet unpublished).

In several places (including Line 387) you conclude that you show an effect of specific immune priming on the evolution of bacterial virulence. Can you really state that given your results that virulence did not change, other than decreasing in both selected lines relative to the ancestral state? I think rewording may be needed.

Finally, this is more minor, but I think more justification for your proxy for “fitness” is needed- you simply note that you combine mortality rates and spore load data (Lines 199-200) but even in the methods, there is no justification for why this is a reasonable proxy for bacterial fitness in this system.

Reviewer #2: (No Response)

Reviewer #3: 1. What prior literature are the authors using to base their hypothesis that immune priming would evolve for higher virulence in Btt? They cite Hoang, et al (2024) and Rafaluk et al (2017) as examples of evolved higher virulence, but they also cite evidence from Ford, et al (2016) that pathogens evolve lower virulence. Mikonranta et al (2015, BMC Ecology and Evolution, https://doi.org/10.1186/s12862-015-0447-5) show that one-side evolution of Serratia marcescens DB11 in D. melanogaster resulted in reduced virulence, which the authors did not cite. It would be helpful to clarify what aspects from the literature motivate the authors hypothesis that Btt evolves higher virulence in response to immune priming.

2. In the second paragraph of the discussion section, the authors discuss three possible explanations to justify the observed lower virulence of evolved Btt lines: 1) evolution of high virulence is costly in the absence of co-evolution, 2) virulence factors encoded on mobile elements like plasmids are costly, 3) virulence evolution could be reduced due to other microbes in the host environment. While the authors do a great job discussing point two, the description of points one and three are minimal and could be expanded on what aspects of their study correspond with these points they cite? Additionally, in point 2, the authors discuss Cry toxins do not play a critical role for Btt virulence in beetles, so why focus on measuring Cry3Aa expression over other virulence factors? Were there any mutations observed in virulence factors like chitinases or proteases, or does plasmid 188kb encode additional virulence genes? Using the existing sequencing results, I think the current study would benefit from the addition of reporting of how other virulence factors were impacted in the evolved lines.

3. The potential increase in lytic phage activity in both evolved lines in Btt is interesting and I think could be expanded on in this study. In the double layer agar experiments, the ancestral strain acts as a negative control since it does not have active phages, so repeating this experiment and the other phenotypes measured (mortality and sporulation) with a Btt strain known to have active phages can be a positive control. Thus, if a Btt line with active phages show similar phenotypes to the evolved strains, potentially, this increased lytic activity maybe a mechanism for reduced virulence. Also, is there a difference in the number of phages between immune primed and non-immune primed evolved strains? The double-layer agar assay is quantifiable, so it should be feasible to score the number of phages to test whether the priming evolved and non-primed evolved strains differ.

4. One of the major findings in this paper is that evolution to immune priming results in increased variation to virulence, however, the authors only show host mortality as the only phenotype to demonstrate this. Increased host mortality is associated with higher virulence phenotypes in pathogens. Since the authors found that Btt evolved in beetle larvae exhibit reduced virulence, I think the study could be complemented by including at least one or two other disease phenotypes associated with virulence. For example, measuring Btt pathogen load during the course of infection or measure pupation rate of larvae that survive acute infection.

**Part III – Minor Issues: Editorial and Data Presentation Modifications**

Reviewer #1: Line 151 – Is variance really the best measure here? See major comments.

Line 159 – can you justify why you used a beta distribution here, versus something like a gamma distribution? I didn’t see justification in the methods either.

Line 173- I suggest deleting the word “where”

Lines 391-393- The sentence “Not only the medical relevance…” is not grammatically complete.

Line 507- there is an error here in the reference source that needs updating.

Reviewer #2: Methods and Figures

Line 152: why is Fig. 2A presented after 2B?

Line 198: the p-value isn’t significant here? The significance threshold was not discussed in the Methods.

Lines 233: Was sequencing done at the population level instead of at the clonal? E.g., the heat map of Fig. 5 suggests allele frequencies within populations. What does this mean in terms of connection to the phenotypic results?

Lines 235 – 236: 32 unique variants?

Line 239: “position 2,017,894” isn’t really general wording. It doesn’t make sense without reading the Methods first.

The data points for Figs. 6a and 7b are not present like they are in previous figures.

Figure 6b: what is the p-value for? what test?

Other comments:

Lines 54 – 55: It is unclear why high virulence would be favored in non-tolerant hosts.

Line 63: what is external host immunity?

Line 65: are they contradictory results? They are just examples of ways that virulence evolutionary trajectories can be altered.

Line 67: Incomplete immunity is not defined

Line 90: I’m guessing this is host and not bacteria, in which case hosts are not evolving so they can’t have generations?

Lines 141 – 144: how is this different than the next paragraph?

Lines 221 – 222: Has cry3a been previously shown to affect virulence of Bt toward the beetle host? Or is this a hypothesis that cry3a is involved and has not been shown before?

Lines 316 – 317: Just a suggestion but it might be better to introduce the coverage data first showing lower cry3aa gene coverage, then go in and test for expression of this gene, and correlation between expression and coverage.

Lines 323 – 325: then why did the authors test this gene?

Lines 327 – 330: it is unclear what the logic is here

Line 331: do beetles have defensive microbes?

Line 372: what specifically about development?

Lines 392 – 393: how is immune priming being used in these fields?

Lines 507 – 508: error

Reviewer #3: Here, I list comments specific to each section of the manuscript.

Introduction:

• More detail on what the authors define as varying levels of virulence would provide welcome context here. For example, line 42 discusses intermediate levels of virulence of the Myxoma virus without describing exactly what intermediate virulence looks like in contrast to high. This could also be revisited in the discussion since the authors did not find evidence for immune primed evolved lines to increase virulence, potentially Btt evolved intermediate virulence mechanisms.

• More information on the activated immune response of primed Tenebrio castaneum vs an active immune response in infected, non-primed larvae would be additional context. For example, description of what are the primary effectors (e.g. antimicrobial peptides, ROS, immune cells) of T. castaneum that are upregulated after infection and priming and which aspect of the response primarily control Btt infection. Additionally, more information on the Btt infection cycle in larvae in the wild would-be helpful context such as including how larvae would encounter spores in nature and how spores reproduce and disperse from larvae cadavers.

• The authors briefly discuss how external host immunity (line 63) and defensive microbes (line 64) are examples of factors that can impact pathogen virulence without much detail on how they define these terms. For clarity, it would be helpful to provide additional context on the type of pathogen and the infecting host from these cited studies is helpful to know because unique properties of different pathogen species and their distinct host environment could influence what virulence strategies they evolve.

• Do the authors consider transmission as a form of virulence? I believe they are connected, but the authors should clarify what makes virulence and transmission distinct.

Results:

“Does host immune priming affect pathogen virulence evolution?” Section

Figure 1:

• What is the motivation or prior work used to justify performing 8 selection cycles, and how many generations of Btt (doubling time) occur during infection? The papers they cite like Hoang, et al (2024) performed 14 selection passages on P. aeruginosa and Ford, et al (2016) passaged Enterococcus faecalis and Staphylococcus aureus for 10 passages.

• For clarity, I think it is helpful to have a prescribed naming convention for the immune primed evolved Btt and the non-primed (author refers to control) evolved Btt to be distinct from the immune primed host environment and the control host environment.

Figure 2:

• Line 142-143: Here the authors are trying to directly compare average mortality between control hosts and primed hosts infected with ancestral, immune primed, and non-immune primed Btt. I think the data across these two figures should be compiled in the same graph if they are going to make direct make the comparisons between the two host environments.

• Line 145: What makes this analysis different than the one performed above? I'd explicitly state what type of model you created in the body of the results and what are the major factors being tested.

• Line 156: The authors state “the primed host, we observed higher variance in virulence across primed-evolved replicates (σ2 = 0.862) than in control evolved replicates (σ2 = 0.147).” What makes these results distinct from what is reported on lines 151-152? From my understanding both sentences are reporting on the same finding that Btt replicates evolved in primed hosts show higher variance in virulence than Btt controls when in a primed host. Since there are statistical comparisons being made in the text that are not present in the main figure texts, the authors should have a table reporting their statistical output.

• Figure 2 presentation:

o Include X axis title for 2A & 2B

o Y axis should include proportion

o The authors should present error bars for their boxplots throughout the text.

Figure S1:

• Line 164-What is the authors motivation for including this figure in the supplement? This seems like a significant finding from their evolution experiment (referenced even in the abstract).

“Do the evolved pathogens differ in spore production, sensitivity to priming and fitness?” Section

• Line 167: I believe the headers for these sections could be better served to state the primary finding from the results section. The questions posed in the headers could be better served to open the motivation for the experiments in each section.

Figure 3:

• Line 174: Swap figure order with Figure 3C since Figure 3D is mentioned first.

• Figure 3 presentation:

o The x and y axis should be a larger font.

o What is the unit of the Y axis? Is it on a log scale?

Figure 4:

• Line 194: In the legend the authors state what virulence is in this case, but it should be reflected in the text as well or more explicitly say host mortality.

• Line 205: The authors state “found a negative trend for the primed-evolved pathogens” when correlating priming sensitivity with pathogen fitness. What does this suggest?

“Do the evolved pathogens differ in 220 Cry gene expression?” Section

• Lines 224-225: When the authors discuss the statistics for the qPCR analysis, there needs to be a transition sentence between the discussion of the analysis that pools all the evolved lines together (lines 223-224) into the statistic performed for each experimental line relative to the ancestral strain.

“Do the evolved pathogens differ in their genomes and mobilome?” Section

• Figure 5 presentation: Need to indicate in figure legend what the darker and lighter colors mean for the heat map.

• Line 238: The authors mention virulent and non-virulent lines when referencing their genomics sequencing. What do the authors mean by this? In the figure 5 legend, they reference Fig. 2 for the order of virulence by evolved lines seen in the plot. The text of the results section does not formally report which of the evolved strains are the most and least virulent, which should be included.

• Line 271: The relative expression of the cry3Aa gene I assume is the qPCR expression measured in the previous section? This should be clarified in the body of the text when presenting Fig. 6B.

Figure S7:

• This supplementary figure needs clear labeling for what each of the plates represent. If possible, zoom in to a portion of the plate so the phage patches are distinct.

Discussion:

• Line 348: I do feel like these are reaches in the discussion and takes away focus from the paper. Could be worth future investigation to explore

• Line 357: Is it a fair comparison to make between pseudomonas and bacillus? Gram-negative vs Gram-positive bacteria could very likely evolve different trajectories since they vary greatly

• Line 361: What are the specific pathogen cues triggering immune priming in their study?

• Line 377: While measuring the host’s immune response to evolved pathogens could be a follow-up study, I think it could be a worthy addition to the paper to compare a more virulent evolved line vs a less virulent evolved line to see how they may impact the hosts immune response. Else, I think this point could be further clarified in the discussion of known examples of pathogens that reduce the immune response in beetle larvae.

Methods:

• Line 489: How long were these cadavers sitting before being homogenized? For the selection experiment above it says the authors waited 7 days. Is that what is happening here?

• Line 560: What do the authors define as plate in this model?

• Line 581: The authors determined Btt fitness in the host by “combining the spore load and proportion of mortality”. How was this performed? Were they summed, made relative to each other? They also mention this in the results section (line 199-200) but do not clarify how they combined these values together.

Grammar:

Below are a few examples in the text that could be rephrased for clarity. Working with a copyeditor could help improve the flow of the text.

• Line 141-142: The authors write ‘Regarding the host environment encountered during the readout, i.e. after evolution had happened’. They use ‘i.e.’ frequently throughout the text, so it could help readability to make sentences like this more succinct.

• Line 398: Sentences starts with “As pathogen”, this can be rephrased.

• Line 507: Possible error with citation manager?

PLOS authors have the option to publish the peer review history of their article (what does this mean? ). If published, this will include your full peer review and any attached files.

**Do you want your identity to be public for this peer review?** For information about this choice, including consent withdrawal, please see our Privacy Policy .

Reviewer #1: No

Reviewer #2: No

Reviewer #3: No

**Figure resubmission:**
---

## [Decision Letter · Decision Letter 1]

7 May 2025

Dear Dr. Korša,

We are pleased to inform you that your manuscript 'Experimental evolution of a pathogen confronted with innate immune memory increases variation in virulence' has been provisionally accepted for publication in PLOS Pathogens.

Best regards,

Robert L. Unckless, Ph.D.

Academic Editor

PLOS Pathogens

Debra Bessen

Section Editor

PLOS Pathogens

Sumita Bhaduri-McIntosh

Editor-in-Chief

PLOS Pathogens

orcid.org/0000-0003-2946-9497

Michael Malim

Editor-in-Chief

PLOS Pathogens

orcid.org/0000-0002-7699-2064

Two expert scientists have reviewed your resubmission "Experimental evolution of a pathogen confronted with innate immune memory increases variation in virulence". Both were enthusiastic about the reviews and the likely impact of the manuscript. Reviewer 1 had a few minor points that should be considered. I look forward to seeing this published.

Reviewer Comments (if any, and for reference):

Reviewer's Responses to Questions

**Part I - Summary**

Reviewer #1: Overall the authors have addressed my major concerns thoroughly.

Reviewer #3: The authors satisfactorily integrated my feedback and that of the other reviewers. I think these greatly improved the paper, so I recommend its acceptance for publication.

**Part II – Major Issues: Key Experiments Required for Acceptance**

Reviewer #1: None

Reviewer #3: (No Response)

**Part III – Minor Issues: Editorial and Data Presentation Modifications**

Reviewer #1: In the second paragraph of the Introduction (Line 73 of the version withOUT track changes), I think it may be a bit hard for readers to follow the logic here without some clarification. Perhaps change the word "actual" to "realized" or something like that? It is always tricky to distinguish between the pathogen's inherent virulence phenotype and the realized virulence in the host, given the hosts background. I also think it would help to add "in hosts with tolerance" to the end of that sentence so that it reads:

"The evolution of tolerance, for instance, can strongly influence the evolution of virulence... without increasing the realized level of virulence IN HOSTS WITH TOLERANCE."

My other editorial comment just relates to the length of that second paragraph. It is quite long and I think could be broken up in 1 or 2 places. Two obvious candidates are:

Line 80 (in version without tracked changes)- new paragraph at "Virulence evolution is a complex trait..."

AND/OR

Line 93- new paragraph at "Invertebrate immune priming has been suggested..."

Reviewer #3: (No Response)

PLOS authors have the option to publish the peer review history of their article (what does this mean? ). If published, this will include your full peer review and any attached files.

**Do you want your identity to be public for this peer review?** For information about this choice, including consent withdrawal, please see our Privacy Policy .

Reviewer #1: No

Reviewer #3: No

---

## [Editor Report · Acceptance letter]

Dear Dr. Korša,

We are delighted to inform you that your manuscript, "Experimental evolution of a pathogen confronted with innate immune memory increases variation in virulence," has been formally accepted for publication in PLOS Pathogens.

Best regards,

Sumita Bhaduri-McIntosh

Editor-in-Chief

PLOS Pathogens

orcid.org/0000-0003-2946-9497

Michael Malim

Editor-in-Chief

PLOS Pathogens

orcid.org/0000-0002-7699-2064